# Study on the Prevention and Control of Downhole Debris Flows Based on Disaster Chain Theory

Xiangdong Niu [1,2,3] , Kepeng Hou [1,2,*] and Huafen Sun [1,2]

1 Faculty of Land and Resources Engineering, Kunming University of Science and Technology, Kunming 650093, China; niuxiangdong@stu.kust.edu.cn (X.N.); 20140091@stu.kust.edu.cn (H.S.)
2 Yunnan Key Laboratory of Sino-German Blue Mining and Utilization of Special Underground Space, Kunming 650093, China
3 Yunnan Yarong Mining Technology Co., Ltd., Kunming 650000, China
* Correspondence: 11301046@stu.kust.edu.cn

**Abstract:** The occurrence of downhole debris flows in caving mines has burst, concealment, and destruction characteristics. This study aimed to investigate accurate prevention and control measures for downhole debris flows. The research background was a downhole debris flow in the Plan copper mine. The disaster chain theory was applied to study prevention and control methods for downhole debris flows. Using a model of source generation, chain breaking, and disaster reduction, we proposed accurate prevention and control measures for downhole debris flow disasters, which prevent and control the downhole debris flows at the source. The results showed that the disaster chain type of downhole debris flow disasters is the compound periodic cycle chain, which has the characteristics of the branch basin chain and the periodic cycle chain. Based on the chain-effect nature of disasters caused by downhole debris flows, active and passive prevention and control methods for downhole debris flow disasters were proposed. The active prevention and control measures for chain breaking and disaster reduction involve isolating the generation conditions from the source, inducing a downhole debris flow disaster. This prevention and control method is difficult to implement during the actual production process. The idea of disaster reduction through passive defensive chain breaking is based on the fact that if the three essential types of conditions for the downhole debris flow formation are not present at the same time, then a disaster accident of a downhole debris flow can be effectively prevented and controlled. Accordingly, the following measures are proposed for preventing and controlling downhole debris flows: (1) reinforcement measures applied to the slope body of the landslide material source in the collapse pit; (2) adopting comprehensive flood control measures such as locking, intercepting, dispersing, draining, and blocking under hydraulic conditions; (3) blocking the formation of the channel by adjusting the ore drawing conditions; (4) addressing the inducing factors by blasting with a small amount of explosive. According to the disaster chain theory, prevention, and control methods for downhole debris flow in caving mines were investigated in this study, which not only broadens the research of the debris flows but also fills the gap in the systematic research on downhole debris flows.

**Keywords:** natural caving mining; downhole debris flow; disaster chain theory; disaster reduction; prevention and control methods

## 1. Introduction

Natural caving is a mining method with high mechanization, good safety, high production efficiency, and low cost. It is the only low-cost underground mining method comparable to open-pit mining. Natural caving has been widely applied in several countries, but the application in China is gradually showing an upward trend. The earth's surface or caving mines with loose materials on the caving ore bed contain fine moraine particles. As the upper overburden of the caving ore bed consists of small particles, the

existence of fine particles in the overburden threatens the safety of mine production and causes unconventional disasters. The Plan copper mine, for example, is a natural caving mine with fine moraine on the surface. As mining depth increases, the area of the surface subsidence expands. During a flood season, downhole debris flow accidents easily occur in the mine because of a covering layer of debris (moraine) several tens of meters thick on the surface of the stope, which mixes with rainwater and flows down the stope through various channels [1–3]. Between 2019 and 2022, more than 10 downhole debris flow accidents occurred in the mine. The high energy of the downhole debris flow burst makes it extremely destructive, which not only restricts the production capacity of the mine but also threatens the mine's safety and efficient production [4–6].

In a narrow sense, a downhole debris flow can be categorized as a mine debris flow. Mine debris flows occur in mountainous areas where mineral resources are concentrated. A debris flow induced by the change in the original topography and geological conditions due to man-made mining is also called a "man-made debris flow" [7–9]. Essential differences exist between mine debris flows and natural surface landslides, tunnel debris flows, and other geological disasters [10,11]. Research related to the metal mine debris flow forecast theory and prevention and control methods is mainly concentrated in the field of open-pit debris flows, but a few reports on the prediction and control of downhole debris flows in downhole metal mines have been published [12–15]. As a downhole debris flow occurs downhole, its occurrence process is often hidden. Consequently, it is difficult to develop prediction, prevention, and control methods for downhole debris flows.

Only a few scholars have investigated this problem. Li et al. [16] used five evaluation factors of total rainfall, daily rainfall, rainfall duration, overburden thickness, and mining depth according to the efficacy coefficient method. They established an early-warning model of a downhole debris flow disaster in the Chengchao iron mine. Chen et al. [17] discussed different stages of mine debris flow prevention and control measures with respect to time. The control of debris flows in mines was divided into three stages: design and construction stage, operation and formation stage, and environment recovery stage. The prevention and control of mine debris flows were investigated by limiting material sources, engineering measures, and environmental control. Zan et al. [18] highlighted the conditions and mechanism of the formation of open-pit debris flows, dump debris flows, and downhole debris flows through a comprehensive literature analysis. Subsequently, the authors proposed preventive, control, management, and technical measures. Ou et al. [19] examined prevention and control methods for downhole debris flow disasters in the Lizhu iron mine and proposed a comprehensive control method based on open-pit and downhole methods. The open pit was treated by removing the debris flow source in the pit, and the adjacent stope passage was blocked by downhole ore. Huang et al. [20] analyzed the geological hazards in the Tianbaoshan mine, Huili, Sichuan province, and proposed fortification and comprehensive control measures. The measures adopted were blocking, filling, dispersing, draining, and protecting; backfilling and dredging are performed before the rainy season each year. Backfilling the subsidence area with waste rock prevents the gathering of torrential rain and flash floods in the area. Chen et al. [21] studied the mechanism of sand bursting under different loose overburden conditions. The results showed that 5% and 10% clay content could promote water production and sand bursting. When the clay content is 20%, the sand body remains stable, and the sand outburst is restrained. Silalahi et al. [22] studied debris flow accidents during mining. They considered the fact that long-distance ore migration could change particle shape and size to form fine and loose particles. The debris flow material is composed of small saturated loose particles and water from the fusion. Xue et al. [23] discussed the causes of mud bursting into a debris flow by summarizing and analyzing the cases of mud bursting in the past several decades. The cause and evolution of mud bursting and the corresponding control methods were introduced.

It is extremely difficult to investigate methods for forecasting, preventing, and controlling downhole debris flows. This difficulty is a result of the uncertainty and complexity

of the interaction between the media of the downhole debris flows and the invisibility of the flow occurrence process. Considering the limited research on this topic, the complexity of the downhole debris flow disasters, and the urgent need to develop prevention and control methods, this study was based on a downhole debris flow in the Plan copper mine. The theory of disaster chain was used to investigate prevention and control methods for downhole debris flows in caving ore mines. Prevention and control measures for downhole debris flow accidents were examined from the perspective of the source and chain breaking. This study contributes to the systematic research on downhole debris flows, improves and broadens the research on mine debris flows, and has practical significance for accurate prevention and control of downhole debris flows.

There are many factors affecting debris flows. When considering these factors, most scholars in the study of debris flow prevention and control methods consider these factors separately and only study the influence of a single factor on the downhole debris flows without taking into account the interaction between various factors. The novelty and innovation of this study are in the consideration of the relationship between many factors affecting the downhole debris flows.

## 2. Overview of the Chain Theory of Catastrophe

The chain theory of catastrophe is a new subject system with a chain effect, which is abstracted from the generality of disasters. The theory posits that material, energy, and information constitute the carriers of disaster, and the model and mechanism of disaster reduction in each stage are studied by analyzing the chain evolution law of the carrier [24–27].

According to the theory, the formation of disasters has a chain regularity. Regardless of the complexity of the factors, different disasters have different patterns and changing situations due to differences in regions, environments, and climates. However, disasters are always formed through a gradual evolutionary process, which reveals that the state of the natural environment evolves in a direction unfavorable to human society, and the process mechanism indicates that the formation of disasters is continuous. The continuous evolution process is always characterized by a certain amount of material, energy, and other information forms, which is the carrier reflection of the disaster chain. This carrier embodies the evolution of the connotation and extension relationship from quantitative changes to qualitative changes. Therefore, the formation process of disasters can be described by the chain relationship or chain effect. The chain form is the abstract of the disaster, and the disaster is the support of the chain form. Since a disaster exists in the form of a chain, if the chain is broken, then the disaster can be effectively avoided along the original chain. This is the ultimate application value of the theoretical research on disaster reduction. The disaster chain carrier reflection block diagram and the evolution block diagram are shown in Figures 1 and 2, respectively.

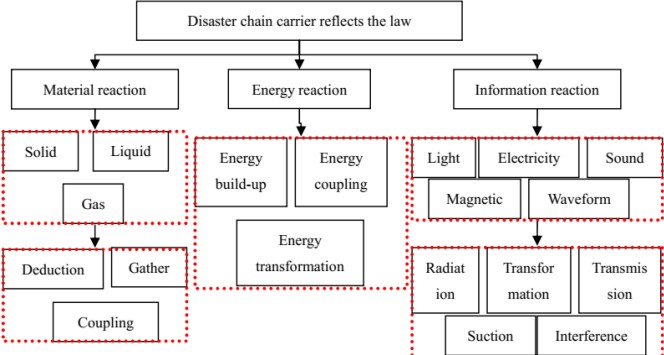

**Figure 1.** Disaster chain carrier reflection block diagram.

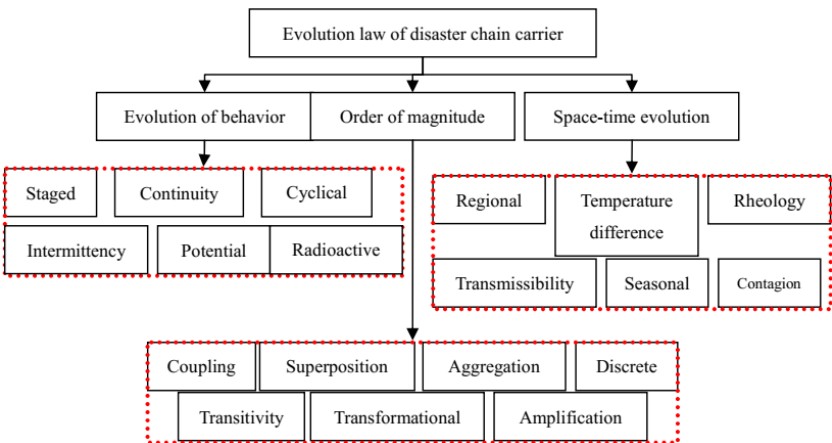

**Figure 2.** Block diagram of disaster chain carrier evolution.

A complete chain-like disaster process includes a disaster-causing ring, trigger ring, damage ring, and chain-breaking ring. The disaster-causing ring is mainly composed of geological factors formed by rock strata and geological structures. The excitation ring is mainly composed of non-geological factors such as man-made and mining disturbances. The damage ring is formed by the disaster loss after the disaster. A broken chain refers to project management and protection measures.

According to the Marxist philosophy, "The development of anything in nature satisfies the general law from birth, development, climax to end," and the occurrence and development of disasters are no exception. The initial stage of a disaster is the incubation stage, that is, the initial stage or incubation stage of the disaster chain. In this stage, the destructive force is considerably weak or has not formed a destructive force, and the information (such as energy) is also in the initial gathering and coupling stage. In this catastrophic process, governance limits the spread and expansion of disasters to the maximum extent. The occurrence of disasters can be reduced by eliminating them in the germination stage, a concept Professor Xiao described as "pregnant chain cut off disaster reduction." Disasters can be classified according to the time interval, stage development characteristics, and degree of damage, as shown in Table 1 [28,29].

**Table 1.** Division of disaster chain stages.

| Stage Division | Phase Characteristics | Destructive Strength | Carrier Information | Time Ratio | Technology Initiatives |
|---|---|---|---|---|---|
| Early stage | Gestation stage | Destructive force has not yet formed | Accumulation of matter and potential energy | Longer, more than 70% | Broken chain |
| Medium stage | Latent stage | Potential for destruction | Storage of matter and potential energy | Ephemeral, about 25% | Defense |
| Late stage | Induction stage | Violent explosion of destruction | Kinetic energy of diffusion of matter bursts forth | Instantaneous, less than 5% | Governance |

## 3. Chain Effect Analysis of Catastrophe

### 3.1. Cases and Their Hazard Characteristics

The Pulang copper mine of Yunnan Diqing Non-ferrous Metal Co., Ltd. (located in Shangri-la, China) is mined by the natural caving method with a production scale

of 12.5 million TPA. Because of the existence of loose particles in ground moraine, the mine easily induces downhole debris flow accidents during the rainy season, which seriously threatens the safety of mine production. Since 2019, the Pulang copper mine has experienced several downhole debris flow accidents, as shown in Figure 3.

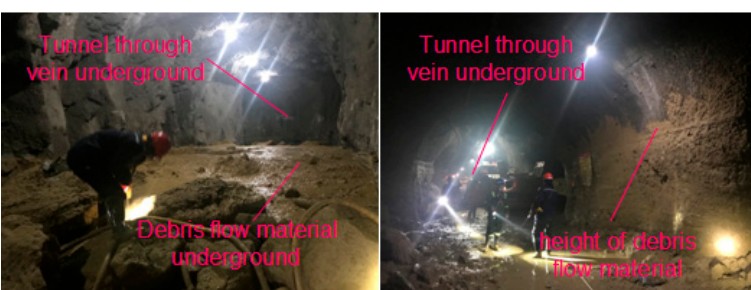

**Figure 3.** Pictures of the debris flow accident in Plan copper mine.

The disaster-causing factors of mine downhole debris flow disasters are complex and from multiple sources because of the different technical conditions and human factors involved. A downhole debris flow is not only a complex two-phase flow but also more complex than a surface debris flow. Therefore, research on the prevention and control of downhole debris flow depends on careful consideration of these complex factors and their dynamic change process to establish their relationship with the disasters.

According to the author's research on the downhole debris flow of the Plan copper mine by natural caving, three essential types of conditions are necessary for the formation of a downhole debris flow: material source conditions, channel conditions, and hydrodynamic conditions. These conditions play an important role in the formation of the debris flow. Figure 4 illustrates the hazard-causing factors of the basic formation conditions of the downhole debris flow in the Plan copper mine.

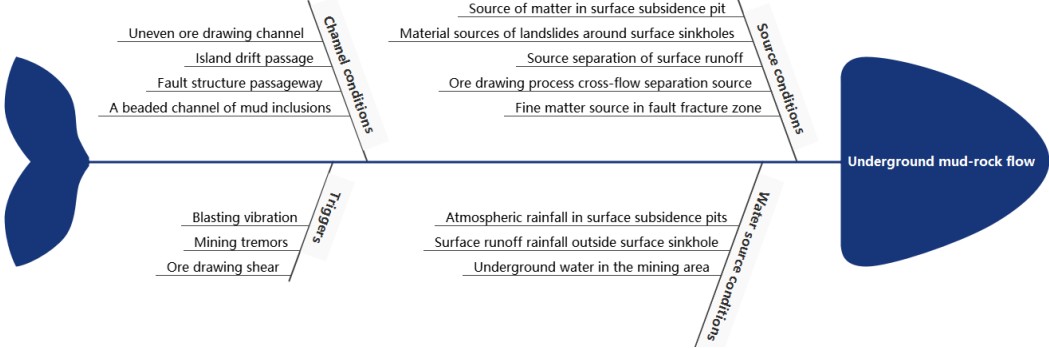

**Figure 4.** Fishbone map of the debris flow hazards in the Plan copper mine.

### 3.2. Chain Type Characteristics

Disaster chains are of different types because of the different causes of disasters or laws of development and evolution. The feature classification of disaster chain type vividly describes the formation mechanism of the disaster chain, the law of morphological evolution, forms of disaster damage and performance, and essential differences among various types of disasters. Depending on the traits reflected by the carrier, the disaster chain can be divided into eight types: collapse–slip chain, periodic cycle chain, branch–basin chain, branch–leaf chain, spreading–erosion chain, scouring–deposition chain, fluctuating attack chain, and radiation-killing chain [30–32]. The main performance of the periodic cycle chain is the periodic reflection of the chain. The periodicity is not the carrier discontinuity of the chain but the peak-valley situation of the chain carrier. The peak period is the period of a strong chain reaction, and the trough period is the period of a potential chain

reaction. Figure 5 shows a schematic of a typical cycle chain. The branch basin chain is composed of several branch systems, which aggregate into the secondary main stream, and the secondary main stream aggregates into the main stream, resulting in the disaster outbreak. The magnitude of the chain increases gradually from a small to a large value and from a weak to a strong state. The number of chains decreases with the aggregation of branches while the destructive energy increases rapidly. Thus, the formation process of the branch-trunk basin chain is the catastrophic step-by-step process of the branch chain confluence source. This process is the process of material, energy, and information, and the destruction intensity increases with the increasing accumulation of destruction, which is a key feature of the chain. Figure 6 shows a schematic of typical tributaries.

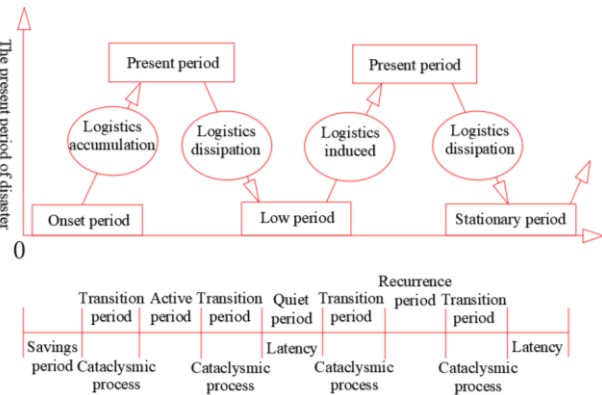

**Figure 5.** Schematic diagram of typical cycle chain.

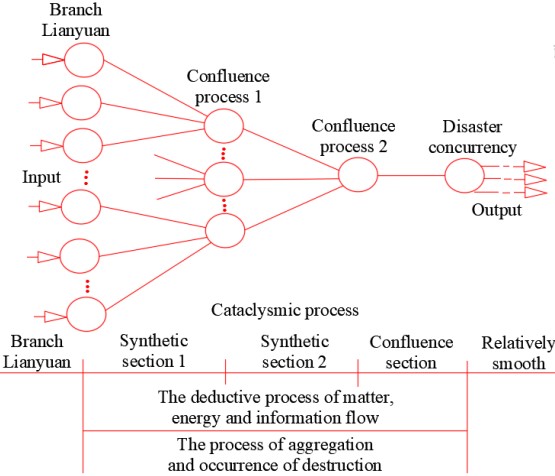

**Figure 6.** Schematic diagram of typical tributaries and watersheds.

The three basic conditions for the formation of debris flow are a material source, terrain (channel), and hydrodynamic conditions. These conditions play an important role in the formation of debris flow. Therefore, the formation of downhole debris flow accidents is a complex process of multiple factors. Each influence factor may lead to the occurrence of a downhole debris flow disaster, but the interaction of multiple factors aggravates the flow formation and shortens the occurrence time, which are characteristics of the branch basin chain. In addition, the Plan copper mine, which is mined by natural caving, is a subtropical alpine monsoon. The rainy season in the mining area is from May to October. The rainfall during the rainy season accounts for 87.1% of the annual rainfall, and the annual average rainfall is 619.9 mm. Because of the annual periodicity of rainfall in the mining area, the occurrence of downhole debris flow also exhibits characteristics of a periodic cycle chain.

In summary, natural caving mine downhole debris flow accidents do not simply correspond to a single disaster chain type. According to the characteristics of mine downhole

debris flow disasters and the catastrophe chain theory, the disaster chain type of natural caving mine downhole debris flow disasters is a compound cycle chain.

### 3.3. Chain Effect of Catastrophe

After the occurrence of the natural caving mine downhole debris flow accident, the disaster chain continues, but more secondary disasters follow. Scholars are primarily concerned with the unpredictability of the occurrence time of the mine downhole debris flow disaster and the uncertainty of the damage degree of the secondary disaster. Figure 7 shows the chain effect diagram of natural caving mine downhole debris flow disasters.

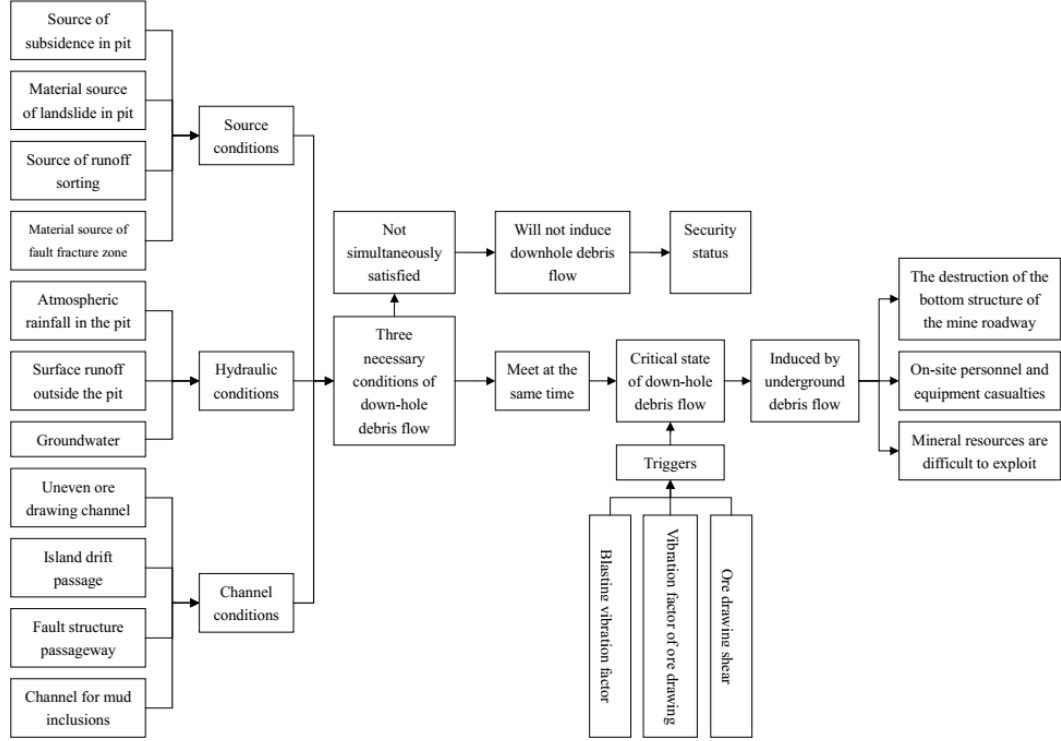

**Figure 7.** Chain effect diagram of debris flow disasters in natural caving mine.

### 4. Prevention and Control of Downhole Debris Flow Disaster

#### 4.1. Active and Passive Controls of Downhole Debris Flow Disaster

In the research on disaster chain control and defense, the "source-broken chain" is a prominent approach. In this approach, the disaster is eliminated at the beginning, and the source of the disaster chain, which is the root of disaster control, is the most effective disaster reduction method. Xiao noted that disaster reduction involves preventing the formation of disaster initiation factors, as well as reducing the occurrence of secondary disasters through the management of single disasters [28,29]. Thus, the starting point of disaster mitigation is not only the earliest source of a single disaster but also the original disaster when the secondary disaster induces itself.

According to the concept of disaster reduction, the active prevention and control measures of natural caving mine downhole debris flow disasters are presented in Figure 8. These measures include cutting off the adverse effects of the source conditions, hydraulic conditions, channel conditions, and inducing factors. However, these factors are difficult to eliminate by technical means in actual situations. For the source conditions of the moraine layer in a cleared or filled surface subsidence pit, during the ore drawing process at the outlet when the mine is in production under the caving method, the fine moraine in the upper part is depleted and penetrates the caving ore bed. In the ore layer, confluence convergence in the bottom structure ore level. At the same time, due to the geological conditions of the fault fragmentation zone debris material, but also the formation of

downhole debris flow to provide the material source conditions, the mine during the rainy season participates in the formation of downhole debris flow hydraulic conditions. The surface runoff rainfall in the watershed outside the sinkhole can be discharged as far as possible through the surface "interception drainage" and other drainage projects but not into the sinkhole. However, the atmospheric rainfall and groundwater in the mining area cannot be eliminated. Regarding the conditions for the formation of downhole debris flow channels, if the uniform drawing pattern is strictly followed during production, the formation of moraine mudflow channels can be avoided. Induced factors of downhole debris flow, such as blasting vibration and ore drawing vibration, can only be reduced as much as mining operations are unaffected. Therefore, the method of cutting off the disaster chain of a downhole debris flow from the source through active prevention and control is theoretically feasible but pragmatically difficult.

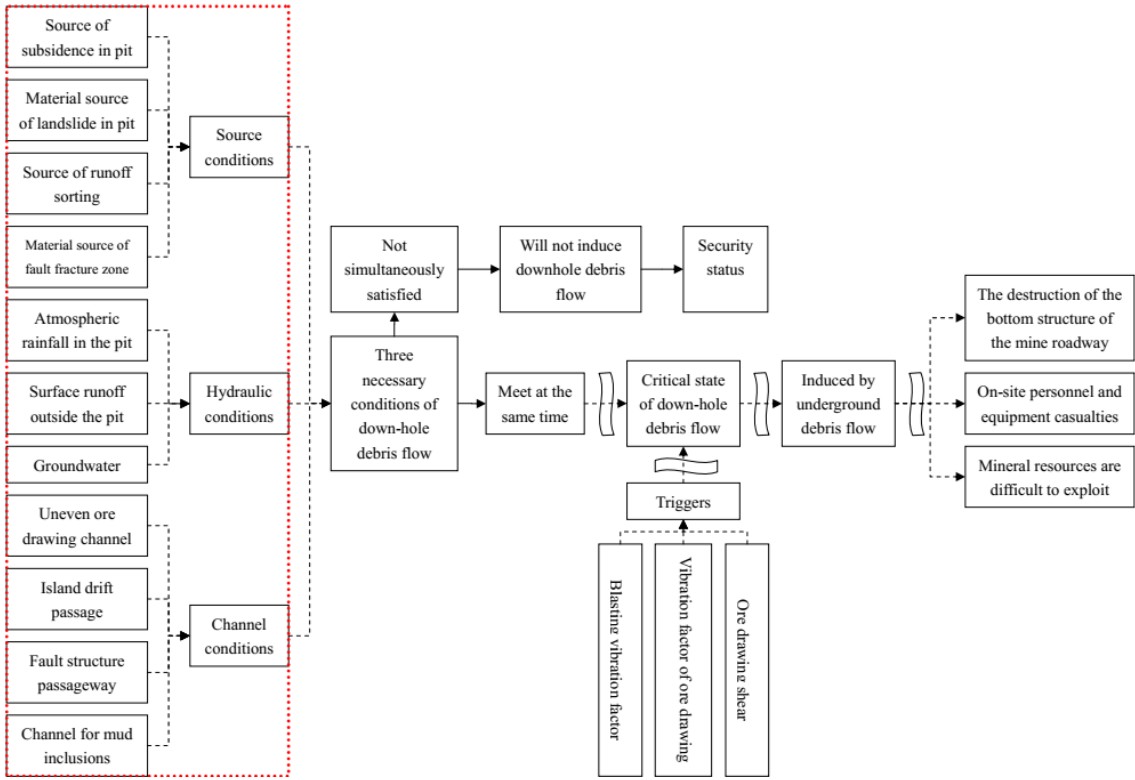

**Figure 8.** Active prevention and control of debris flow in Plan copper mine.

To eliminate or reduce the threat of harm from downhole debris flow disasters to downhole workers or equipment, it is necessary to study other ways of breaking the disaster chain. Figure 9 is based on the proposed passive defense-type program, by which it is relatively easy to implement prevention and control measures. According to Figure 9, the passive defensive chain-breaking approach to reducing downhole debris flow disasters does not cut off the production conditions from the disaster source. The approach is rather based on three necessary conditions for the formation of downhole debris flow. Cutting off one of the three conditions can stop the formation of downhole debris flow, preventing and controlling the downhole debris flow.

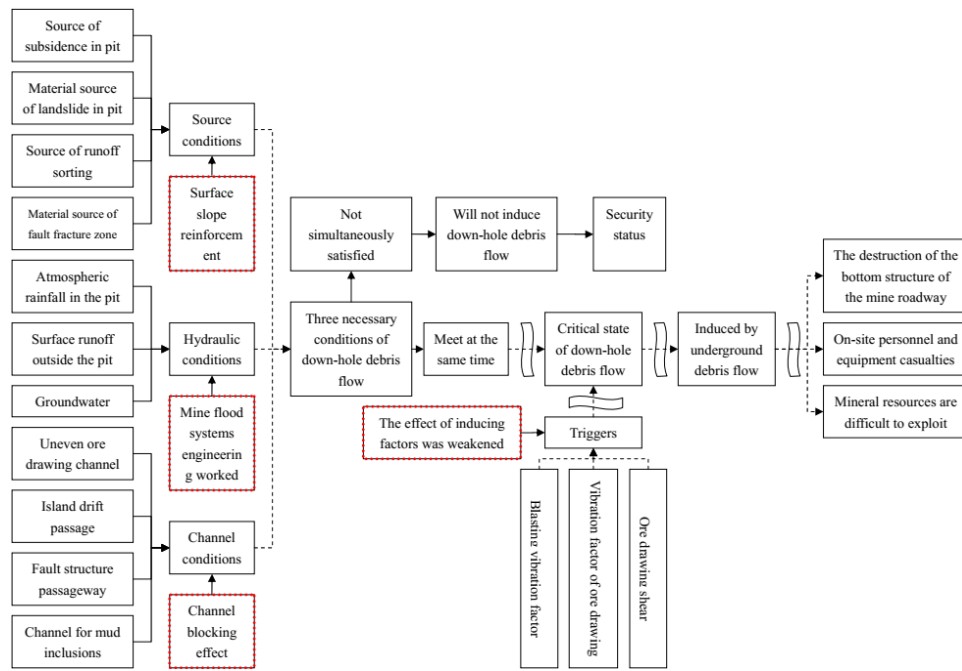

**Figure 9.** Passive defense and control of debris flow in Plan copper mine.

### 4.2. Prevention and Control Measures of Downhole Debris Flow

For natural caving mine downhole debris flow accidents, knowing the scale and time ahead would enable timely planning and implementation of response measures. Thus, the harm caused by downhole debris flow accidents can be completely avoided and controlled. According to Figures 8 and 9, active and passive prevention and control methods can be determined. Compared with the active prevention and control method, the passive prevention and mitigation scheme does not aim to eradicate the three necessary conditions and inducing factors of the formation of downhole debris flow. The scheme rather aims to control the formation of the three conditions to ensure that they are met at the same time. Subsequently, the disaster of the downhole debris flow can be contained for effective prevention and control. The following prevention and control measures can thus be taken:

(1) Prevention and control measures of material source conditions: In view of the landslide material source in the surface subsidence pit, some measures such as slope body reinforcement can be adopted to prevent the slope body from sliding down and the fine moraine material from entering the subsidence, thus, the formation of a downhole debris flow due to the material source conditions is avoided;

(2) Prevention and control measures for hydraulic conditions: Water is an important factor in the occurrence of a debris flow, and the entrance of water into the mine subsidence pit mainly occurs in two ways: atmospheric rainfall around the subsidence pit and surface runoff of rainfall around the subsidence pit. Water in the sinkhole mainly results from surface runoff. Therefore, according to the hydraulic conditions of the prevention and control measures for surface runoff, "interception, drainage, blocking," and other comprehensive prevention and control measures should be taken. The surface runoff of rainwater outside the sinkhole is collected and channeled outside the sinkhole for discharge. Hence, the surface runoff of rainwater upstream of the sinkhole is effectively intercepted, and the surface runoff of rainwater is prevented from entering the sinkhole as much as possible, as shown in Figure 10;

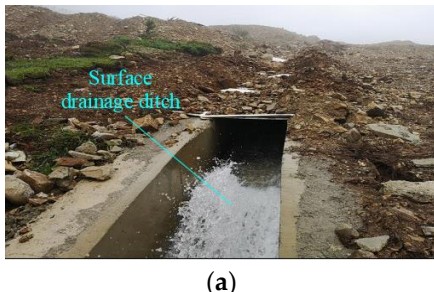 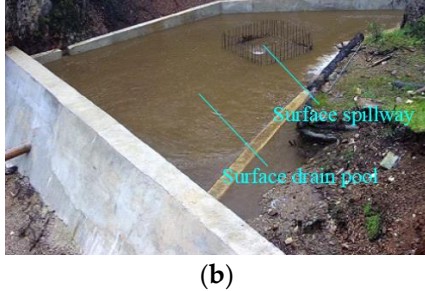

(**a**)          (**b**)

**Figure 10.** Surface flood control and drainage systems within the Plan mine sinkhole. (**a**) Surface flood control by intercepting ditch; (**b**) surface spillway.

(3) Prevention and control measures for channel conditions: When the fine moraine particles overlying the ore layer undergo uneven drawing, a moraine flow channel is easily formed in the caving ore layer, as shown in Figure 11. If the debris flow channel is formed in the caving ore bed because of various reasons during ore drawing, the channel can be blocked by adjusting the ore drawing mode.

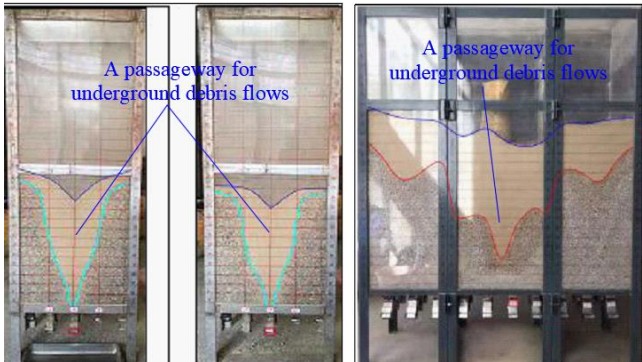

**Figure 11.** Moraine logistics channel under indoor unbalanced ore drawing conditions.

(4) Prevention and control of triggers: The inducing factors play a key role in the downhole debris flow formation. To reduce the influence of inducing factors on the downhole debris flow, the mine adopts the method of multiple blasting with a small number of explosives during the rainy season to reduce the influence of blast vibrations.

*4.3. Evaluation of Prevention and Control Effect*

In 2019, the Plan copper mine started experiencing downhole debris flow accidents. The author conducted a systematic study on the formation mechanism and prevention and control measures of downhole debris flow accidents in the mine. Since 2020, the research results of the Evaluation Standard and the division prevention and control of downhole debris flows have been applied to the prevention and control of downhole debris flows in the Plan mine. In 2019, six downhole debris flow accidents occurred prior to the study of downhole debris flows. Since 2020, the research results have been applied to the prevention and control of downhole debris flows, deepening the understanding of downhole debris flows. Both the 2021 mine and the Plan mine in 2022 had a downhole debris flow accident that year. Figure 12 illustrates the frequency and reduction rate of downhole debris flows in the Plan copper mine during the rainy season from 2019 to 2022.

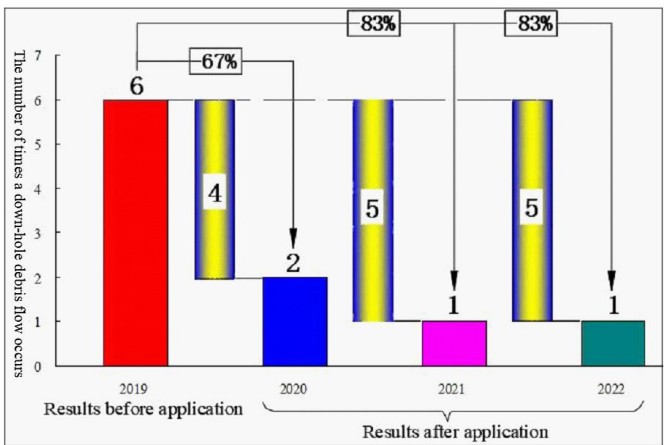

**Figure 12.** Frequency and reduction rate of downhole debris flow in Plan copper mine from 2019 to 2022.

According to Figure 12, the method proposed in this paper for preventing and controlling downhole debris flows has been successful in the prevention and control of downhole debris flows by natural caving. Meanwhile, the number of downhole debris flows in the Plan mine decreased considerably in the past years, which also verifies the correctness and reliability of the prevention and control approach of downhole debris flows by natural caving.

## 5. Conclusions

The downhole debris flow in the Plan copper mine was taken as a research background in this study. The theory of disaster chain was applied to study prevention and control methods for the downhole debris flow in the caving mine from the perspective of source generation, chain breaking, and disaster reduction. The following conclusions were drawn:

(1) The formation of downhole debris flow accidents is a complex process resulting from multiple factors. Each influencing factor may lead to the occurrence of a downhole debris flow disaster, and the interaction of multiple factors aggravates the flow formation and shortens the disaster occurrence time. Downhole debris flow accidents have the characteristics of the tributary basin chain and periodic cycle chain. According to the characteristics of mine downhole debris flow disasters and the catastrophe chain theory, the disaster chain type of the natural caving mine downhole debris flow disasters is the compound cycle chain;

(2) Taking into account the natural caving method of the Plan copper mine, the formation of the downhole debris flow requires three essential types of conditions: source conditions, channel conditions, and hydrodynamic conditions. The disaster chain–effect relationship of the disaster-causing factors in a downhole debris flow disaster in a natural caving mine is presented. Thus, this paper proposes control methods of active prevention and passive prevention of downhole debris flow disasters. The active prevention and control measures of chain breaking and disaster reduction involve cutting off the generation conditions from the source inducing the downhole debris flow disaster, and the measures are difficult to implement in ideal prevention and control scenarios of the actual production process. From the perspective of the three basic conditions for the formation of downhole debris flows, only cutting off one of the conditions can stop the formation of downhole debris flows, preventing and controlling downhole debris flows;

(3) The idea of passive defensive chain-breaking disaster reduction does not aim to eradicate the three essential conditions and inducing factors of the formation of downhole debris flows. However, the idea is to control the formation of the three essential conditions that induce the formation of the downhole debris flow such that the three conditions are met simultaneously. Thus, the downhole debris flow disaster can be effectively prevented and controlled. According to the idea of passive defensive chain-breaking disaster reduc-

tion, the following preventive and control measures should be taken: (1) material source condition adopts the reinforcement measures to the landslide material source slope body in the collapse pit; (2) comprehensive flood control measures such as locking, intercepting, dispersing, draining and blocking should be adopted under hydraulic conditions; (3) the formation of the channel should be blocked by adjusting the ore drawing conditions; (4) induced factors should be considered by using a small amount of explosive for blasting;

(4) According to the occurrence frequency and reduction rate of downhole debris flows during the rainy season from 2019 to 2022 in the Plan copper mine, the proposed prevention and control method for downhole debris flows was successfully applied to the prevention and control of downhole debris flows in a natural caving mine. Meanwhile, the number of downhole debris flows in the Plan mine was considerably reduced, which verifies the correctness and reliability of the proposed prevention and control method.

## 6. Future Development of This Study

Natural caving is a mining method with high mechanization, good safety, high production efficiency, and low cost. It is the only low-cost downhole mining method comparable to an open pit. Currently, natural caving is widely used in other countries, but the domestic application also shows a rising trend. In the process of natural caving mining, with the continuous extraction of downhole mineral resources, the downhole space increases gradually, which inevitably leads to surface subsidence deformation and subsidence. With the increase in mining depth, more frequent debris flow disasters occur as mining progresses. Therefore, the results of this study can provide guidance for the accurate prevention and control of downhole debris flows. The results of this study also have high application prospects in the future.

**Author Contributions:** Conceptualization, X.N. and K.H.; methodology, K.H.; software, X.N. and K.H.; validation, K.H. and H.S.; writing—original draft preparation, K.H. and H.S. All authors have read and agreed to the published version of the manuscript.

**Funding:** This study receives no financial support.

**Data Availability Statement:** The basic data supporting the research results are all in the article.

**Acknowledgments:** The authors would like to thank all the reviewers for providing English editing services during the preparation of this manuscript.

**Conflicts of Interest:** The authors declare that they have no known competing financial interests or personal relationships that could have appeared to influence the work reported in this paper.

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
