# Peer review of "Study on the Prevention and Control of Downhole Debris Flows Based on Disaster Chain Theory"

_water, doi:10.3390/w15132367_

Round 1
Reviewer 1 Report
My comments as below. With the complexity of underground mining is becoming more and more difficult, at present underground debris flow accidents are more and more frequent. At present, it is difficult to prevent and control down-hole debris flow accidents because of its concealment and unpredictability. In this paper, the concept of “Disaster chain theory” is proposed to prevent and control underground debris flow in underground mines, which is very innovative and of practical significance in the field. This paper is a fine, but there are still some things that need to be right.
1. Debris flow is the result of the coupling of material source, channel and rainfall. There are a lot of factors to induce it, but it needs to have three basic conditions at the same time. Therefore, as long as the formation of debris flow control of the three conditions are not at the same time, we can achieve its prevention and control. In this paper, based on the “Disaster chain theory” in the “Source-broken chain” approach to debris flow prevention and control, that the proposed control idea is correct, very much agree with the point of view, this is a very valuable research paper.
2. There are some grammatical errors in this paper, which need to be corrected and corrected, and the author is suggested to polish them if necessary.
3. In the“1 introduction” of this paper, what are the advantages of supplementary natural caving method over other mining methods.
4. This paper is based on the research background of the underground debris flow in Plan copper mine, please add the example of the underground debris flow in pulang copper mine.
5. Figure 8 in the paper, the material source of landslide in the material source condition can be realized by reinforcing the surface slope. It is suggested that the author should complement and perfect it to make the paper more rigorous.
6. According to Article 4, in the article“4.2 prevention and control measures”, supplement the source conditions of the prevention and control measures and methods.
7. In the part of “References”, the reference [5] is incomplete and needs to be improved. In addition, other references need to be standardized and complete.
Needs to be checked some spell mistakes.
Author Response
My comments as below. With the complexity of underground mining is becoming more and more difficult, at present underground debris flow accidents are more and more frequent. At present, it is difficult to prevent and control down-hole debris flow accidents because of its concealment and unpredictability. In this paper, the concept of “Disaster chain theory” is proposed to prevent and control underground debris flow in underground mines, which is very innovative and of practical significance in the field. This paper is a fine, but there are still some things that need to be right.
Point 1: Debris flow is the result of the coupling of material source, channel and rainfall. There are a lot of factors to induce it, but it needs to have three basic conditions at the same time. Therefore, as long as the formation of debris flow control of the three conditions are not at the same time, we can achieve its prevention and control. In this paper, based on the “Disaster chain theory” in the “Source-broken chain” approach to debris flow prevention and control, that the proposed control idea is correct, very much agree with the point of view, this is a very valuable research paper.
Response 1: First, I would like to thank reviewer 1 for his recognition of this thesis. Based on the disaster chain theory, the prevention and control methods of down-hole debris flow are studied. From the point of origin and chain breaking disaster reduction, this paper puts forward the accurate prevention and control measures of the down-hole debris flow disaster, and realizes the prevention and control of the down-hole debris flow from the source, and pointed out the prevention and control measures of underground debris flow.
Point 2: There are some grammatical errors in this paper, which need to be corrected and corrected, and the author is suggested to polish them if necessary.
Response 2: In view of the existence of some grammar errors in this article, the author has carried on the careful examination and the correction to it. In order to ensure the quality of the paper, the author polished the paper. The revised contents are detailed in the red font, and the author has also revised and improved the references according to the periodical requirements.
Point 3: In the“1 introduction” of this paper, what are the advantages of supplementary natural caving method over other mining methods.
Response 3: The advantages of natural caving over other mining methods are: Natural caving is a mining method with high mechanization, good safety, high production efficiency, and a low cost. It is the only low-cost underground mining method comparable to open-pit mining. The author has already made a supplement in the“1 introduction” of this article, the specific supplementary content is detailed in the red mark font.
Point 4: This paper is based on the research background of the underground debris flow in Plan copper mine, please add the example of the underground debris flow in pulang copper mine.
Response 4: In “Section 4.2”, the author has provided examples of landslides in the Plan mine, detailed in the red text.
Point 5: Figure 8 in the paper, the material source of landslide in the material source condition can be realized by reinforcing the surface slope. It is suggested that the author should complement and perfect it to make the paper more rigorous.
Response 5: The author has added “Slope reinforcement measures” in Figure 8 of the paper and “Slope reinforcement measures” in “Prevention and control measures in Section 4.2”. Please refer to the red font section of the article for details of the changes and additions.
Point 6: According to Article 5, in the article“4.2 prevention and control measures”, supplement the source conditions of the prevention and control measures and methods.
Response 6: The author adds “Slope reinforcement measures” in “Section 4.2 prevention and control measures”. Please refer to the red font in the article for the specific additions and additions.
Point 7: In the part of “References”, the reference [5] is incomplete and needs to be improved. In addition, other references need to be standardized and complete.
Response 7: The author has supplemented and improved the incomplete literature [5] . In addition, other references in this paper are standardized and integrated according to the requirements of journals.
Thank you again for your time and consideration.
Wish you all the best.
Sincerely yours,
Xiangdong Niu
2023-06-13
Tel:+86 15198934646
E-mail: niuxiangdong@stu.kust.edu.cn

Reviewer 2 Report
This paper describes a study on the prevention and control of debris flows based on the the disaster chain theory.
Although the topic addressed in this paper can be of interest for the readers of the journal, in my opinion the submitted paper is not appropriate for the publication in the present form for several reasons:
· It is not clear which is the main topic of the paper and how the disaster chain theory has been applied in this study for the prevention and the mitigation of the debris flows risk.
· There is not a comprehensive literature review aimed at adequately presenting the background and what is the novelty of this study in relation to the state of the art. The authors must carry out an extensive literature review including all the relevant papers published recently in international journals. In addition, the references in the main text and in the list of the references do not match the journal guidelines.
· The present structure of the paper does not match the requirements of a reseach paper that should show: the methodology, the case study, the main results and the possible future developments of this study.
The quality of the English is not appropriated: there are many typos and errors and the paper is difficult to follow in many parts.
Author Response
This paper describes a study on the prevention and control of debris flows based on the disaster chain theory.
Although the topic addressed in this paper can be of interest for the readers of the journal, in my opinion the submitted paper is not appropriate for the publication in the present form for several reasons:
Point 1: It is not clear which is the main topic of the paper and how the disaster chain theory has been applied in this study for the prevention and the mitigation of the debris flows risk.
Response 1: Dear reviewer 2, the author would like to express his deep apology to you for the poor experience that may have been caused by the quality of the English paper. Due to the quality of the paper English problems, resulting in you do not fully understand the theme of the paper. The main theme of this paper is: the method of disaster chain theory is used to study the prevention and control of underground debris flow. From the angle of source-generating, chain-breaking and disaster-reducing, this paper puts forward the accurate prevention and control measures of downhole debris flow disasters, which can realize the prevention and control of downhole debris flow from the source.
Underground debris flow is not only a complex two-phase flow. Compared with surface debris flow, underground debris flow has more influence factors, and the relationship among them is more complicated. It is difficult to prevent and control the downhole debris flow when only one or some of the influencing factors are considered in the research. The main technical difficulty of preventing and controlling downhole debris flow is that there are many factors that affect the formation of downhole debris flow, and there are interactions among them. Most importantly, there is a chain effect relationship among the influencing factors. Therefore, based on the disaster chain theory, this paper studies the prevention and control methods of downhole debris flow. By drawing the chain-broken disaster reduction map of underground debris flow, the precise prevention and control measures of underground debris flow are put forward, to reduce the probability and risk of downhole debris flow.
Point 2: There is not a comprehensive literature review aimed at adequately presenting the background and what is the novelty of this study in relation to the state of the art. The authors must carry out an extensive literature review including all the relevant papers published recently in international journals. In addition, the references in the main text and in the list of the references do not match the journal guidelines.
Response 2: There are essential differences between the occurrence mechanism and prevention and control methods of mine debris flow and natural surface landslide debris flow. At present, the research on prediction and prevention and control of underground debris flow mainly focuses on open-pit debris flow, but there are few reports on prediction and prevention and control of underground debris flow. Because of the underground debris flow, its occurrence process is often hidden, which makes it difficult to predict and control the debris flow. At present, only a few scholars at home and abroad have made relevant research on it, the author has introduced and summarized it, see the “Introduction 1” in detail. The novelty of this study compared with the existing technology is reflected in: the existing research results show that most scholars in the study of debris flow prevention and control methods, although there are many factors affecting debris flow, however, when considering these factors, we consider them separately, and only consider the influence of single factor on the downhole debris flow, without considering the interaction between the various factors. The novelty and innovation of this study is to consider the relationship between the many factors affecting the downhole debris flow. In addition, the author has perfected and standardized the text and references in accordance with the standards and requirements of journals.
Point 3: The present structure of the paper does not match the requirements of a reseach paper that should show: the methodology, the case study, the main results and the possible future developments of this study.
Response 3: The author has adjusted the structure of the paper. Chapter 2 and Chapter 3 introduce the methodology and case, chapter 4 introduces the prevention and control methods, and Chapter 5 introduces the main results, added Chapter 6 of the research results of the future development.
Point 4: The quality of the English is not appropriated: there are many typos and errors and the paper is difficult to follow in many parts.
Response 4: In view of the existence of some grammar errors in this article, the author has carried on the careful examination and the correction to it. In order to ensure the quality of the paper, the author polished the paper. The revised contents are detailed in the red font, and the author has also revised and improved the references according to the periodical requirements.
Thank you again for your time and consideration.
Wish you all the best.
Sincerely yours,
Xiangdong Niu
2023-06-13
Tel:+86 15198934646
E-mail: niuxiangdong@stu.kust.edu.cn

Round 2
Reviewer 2 Report
Although the quality and the novelty of the paper are still questionable. I recognize the huge effort of the authors for improving the manuscript and their attempt to answer to my comments. Thus, this version of the paper can be evaluated by the editor for the publication.
A professional editing of the English is required in my opinion.